# Determining the Genetic Control of Common Bean Early-Growth Rate Using Unmanned Aerial Vehicles

**Travis A. Parker, Antonia Palkovic and Paul Gepts *** 

Department of Plant Sciences/MS1, Section of Crop & Ecosystem Sciences, University of California,
1 Shields Avenue, Davis, CA 95616-8780, USA; trparker@ucdavis.edu (T.A.P.); alpalkovic@ucdavis.edu (A.P.)
* Correspondence: plgepts@ucdavis.edu; Tel.: +1-530-752-7743

**Abstract:** Vigorous early-season growth rate allows crops to compete more effectively against weeds and to conserve soil moisture in arid areas. These traits are of increasing economic importance due to changing consumer demand, reduced labor availability, and climate-change-related increasing global aridity. Many crop species, including common bean, show genetic variation in growth rate, between varieties. Despite this, the genetic basis of early-season growth has not been well-resolved in the species, in part due to historic phenotyping challenges. Using a range of UAV- and ground-based methods, we evaluated the early-season growth vigor of two populations. These growth data were used to find genetic regions associated with several growth parameters. Our results suggest that early-season growth rate is the result of complex interactions between several genetic and environmental factors. They also highlight the need for high-precision phenotyping provided by UAVs. The quantitative trait loci (QTLs) identified in this study are the first in common bean to be identified remotely using UAV technology. These will be useful for developing crop varieties that compete with weeds and use water more effectively. Ultimately, this will improve crop productivity in the face of changing climatic conditions and will mitigate the need for water and resource-intensive forms of weed control.

**Keywords:** growth rate; vigor; UAV; sUAS; organic; weed competitiveness; *Phaseolus vulgaris*

## 1. Introduction

Weeds are responsible for worldwide yield losses approximately equal to those incurred by all other pathogens and animal pests combined [1], and their control is a major economic and ecological burden. The competition between crops and weeds operates as a positive feedback loop, and advantages in resource capture early in the season, allow one competitor to outcompete its neighbors for further resources. The strong positive effect of early-season growth rate on crop competitiveness against weeds, has been studied in many major crops [2–5]. These studies demonstrate that a fast growth rate in both canopy area and canopy height are positively associated with weed competitiveness [6]. Consumers are also increasingly interested in forms of agriculture with reduced use of fossil-fuel intensive inputs [7]. Organic agriculture, for example, forbids the use of synthetic fertilizer and herbicides, and experienced 9.7% average annual growth from 1999–2017 [8]. Climate change might also lead to environments that facilitate herbicide resistance in weeds [9]. Rapid early-season growth also shades and insulates the soil, reducing the amount of soil water wasted to the atmosphere, due to evaporation [10,11]. This is of critical importance for crop climate resilience, as future decades are predicted to be increasingly arid [12]. Breeding to improve early season growth rate would improve crop productivity and efficiency with relatively few drawbacks, but the trait is poorly understood genetically in most crops.

In soybean, a grain legume grown globally for its oil- and protein-rich seeds, above-ground plant canopy area and canopy height are correlated with weed suppressive ability during the critical periods for weed control [3,13–16]. This vigorous early-season growth is critical for managing weeds,

while minimizing the use of herbicides and manual weeding. Average canopy coverage is also positively correlated with yield in soybean, therefore indicating that improving the trait might have a positive effect on the final seed yield [17]. The effectiveness of improving yields through selection for the correlated trait of greater canopy coverage alone has recently been demonstrated [18].

Despite the clear need for improved early-season growth rate across a range of crops, difficulties in evaluating the trait have been a major hindrance to its improvement. Traditionally, early-season growth vigor has been evaluated either visually using low-precision, high-throughput subjective methods, e.g., [19] or by a variety of high-precision, low-throughput methods, e.g., [20]. These methods have been insufficient for determining precise values related to growth, while simultaneously being scalable to the level of large populations. Finally, environmental variables have a strong influence on early-season growth rate, so evaluations must be conducted over multiple years and field sites. This often translates into large costs associated with selecting for these traits and studying the genetic causes for their variation. Improved methods would be extremely useful for the evaluation of early-season growth rate.

The use of Unmanned Aerial Vehicles (UAVs, also known as sUAS or drones) has dramatically increased in recent years, due to improved technological and regulatory developments [21]. These technological developments include improved mission planning software, superior aircraft performance, inexpensive high-precision sensors, and improved photogrammetry. Changes in UAV regulations, including part 107 remote pilot certification in the United States, has also made the use of UAVs more practical. Unlike traditional phenotyping approaches, the workload required for UAV-based phenotyping does not scale as a linear function of the number of plots in a trial. These methods are therefore well-suited to large-scale evaluations targeting weak heritable traits, such as growth rate. Previous UAV-based evaluations of soybean have successfully identified a major quantitative trait locus (QTL) for canopy cover on Gm19 [17], which contains the *Dt1* allele for determinacy [22], although the two might not be precisely collocated. The use of UAV-based phenotyping methods will continue to grow as a major asset for plant genetics and breeding in the 21st century.

Common bean (*Phaseolus vulgaris* L.) is a nitrogen-fixing grain legume that has low fertilization requirements, breaks up pest and disease cycles for other crops, and improves soil structure. The species is the primary legume for direct human consumption and is a major source of nutrition for hundreds of millions of people globally [23,24]. Consumption rates are highest in developing countries of the global south, where it serves as a major source of protein and micronutrition. In developed countries, there have been tremendous increases in dry bean production in the organic sector [25]. Organic agriculture is particularly limited in herbicide use. Organic growers often spend 2–5 times the amount on manual hand-weeding than their conventional counterparts [26], and there is a strong demand for alternative means of control, such as through improved early-season vigor of the crop plant. Common bean varieties show variation in early-season growth rate. A more thorough understanding of this genetic diversity would be useful for improving weed competitiveness in the species.

In this work, we have sought to apply advances in UAV technology to identify the genetic basis of vigorous early-growth in common bean. This work was conducted on multiple growth-related traits, including canopy area, height, and volume. It was also conducted across multiple populations, including a diversity panel that represents a large range of phenotypes seen in the domesticated gene pool, as well as a recombinant inbred (RI) population bred ad hoc, to study growth rate. Our results would be valuable for breeding new crop varieties that are productive and resource-efficient in the face of climate change. They also shed light on the expanding utility of UAV-based remotesensing technologies.

## 2. Material and Methods

### 2.1. Plant Materials

The genetic basis of early-season growth rate was evaluated in two distinct populations. The first was the BeanCAP Middle American Diversity Panel (MDP, [27]). This population included 280 varieties that descended from the Middle American domestication event of *Phaseolus vulgaris* [28,29]. These varieties

were primarily indeterminate, with only 19 exceptions that had a determinate growth habit (7%; [30]). This reduced any potential confounding effect of the large-impact but well-studied *PvTFL1y* and *PvTFL1z* mutations related to determinacy [31,32]. In 2016, the population grew in an unreplicated augmented design (Figure S1, [33]) on transitional organic land in Sutter County, California (38.80° Lat., −121.64° Long.). In 2017 and 2018, two replicates of each variety in the population were grown on conventionally managed ground at the Plant Sciences Field Facility, UC Davis (38.53°, −121.78°). Seeds of all populations in all years were inoculated with Gard-N nitrogen fixing bacteria, before planting.

A second population of 207 RILs was developed through reciprocal crosses between cultivar 'Black Nightfall' (W6 51267) and cultivar 'Orca' (PI 632344) [34]. Preliminary screenings of highly diverse heirloom and elite varieties in 2014 and 2015 had determined that Black Nightfall exhibited a more rapid growth rate than Orca. Both of these varieties were indeterminate and Middle American in ancestry. Black Nightfall displayed a type IIIB growth habit, with a large, viny, and somewhat prostrate canopy [30]. In contrast, Orca displayed a small indeterminate type IIA growth habit [30]. This population ("BxO") was grown in an augmented design, each year for three years. Black Nightfall and Orca were among the controls for each block in the population. In 2017 and 2018, 30 seeds were planted in double-row 5 ft × 5 ft plots. In 2019, 120 seeds were planted in double-row 20 ft × 5 ft plots. All trials using this population were conducted on conventionally managed grounds at the Plant Sciences Field Facility of UC Davis (38.53° Lat., −121.78° Long.).

## 2.2. Phenotyping

Both conventional and UAV-based methods (Figure 1) were used to evaluate the growth parameters. The number of plants emerging was hand-counted in each plot, each year, to correct for the effect of the germination rate on the plot growth rate metrics. UAV flights to evaluate growth rate were then conducted at 21, 28, 35, and 42 days after planting (DAP). This period represented a time of rapid canopy development in the common bean. Starting around 42 DAP, plants shifted resources into extensive reproductive development, producing flowers, fruits, and seeds. Continued strong vegetative growth after this point might come at the cost of reproductive investment and is not clearly advantageous.

At each weekly evaluation, a UAV was flown over the field at an altitude of 17–20 m. In 2016 and 2017, RGB imagery was captured using a DJI Phantom 3 12 MP stock camera with a Sony EXMOR 1/2.3 sensor. In 2017, multispectral imagery was collected using a Parrot Sequoia, and in 2018 and 2019 a Micasense RedEdge-M was used for multispectral imagery acquisition (Table 1). Between 2016 and 2017, the camera used was updated to provide calibrated imagery and to add a near infrared band, which allows for the calculation of NDVI and potentially improves the ability to distinguish plants from non-plants in classification rasters. Between 2017 and 2018, the hardware was updated to improve the hardware system reliability and ground sampling distance (GSD). The GSD for all imagery was below 2 cm/pixel (Table 1), whereas plant canopies were on the scale of square meters. RGB imagery was taken on all flight dates, including those where the multispectral cameras were the primary sensor, as a backup and as a reference for data quality. All images were captured automatically using the Map Pilot (2016), Atlas Flight (2017), and DJI Ground Station Pro (2018, 2019), for mission planning, based on hardware compatibility (Table 1).

Raw images from each flight were built into field-scale orthomosaics and digital surface models (DSMs) using Pix4Dmapper version 4.3.31. True-color orthomosaics were generated from RGB imagery and normalized difference vegetation index (NDVI, [35]) orthomosaics were generated from the multispectral imagery. Processed orthomosaics and DSMs were loaded into the Quantum GIS ("QGIS") version 2.18 ([36]). The excess green (ExG, [37]) vegetation index was calculated from the true-color orthomosaics, using the QGIS raster calculator function. ExG and NDVI rasters were then used for the threshold-based image classification to distinguish canopy from soil. These classification layers were used to make canopy-specific DSMs (cDSM, soil raster values filtered) and soil-specific DSMs (sDSM, canopy raster values filtered). All calculations and measurements were conducted within a flight sampling date, however, the orthomosaics between the sampling dates aligned well (<1 m discrepancy)

and without apparent distortions. The QGIS vector grid function was used as a high-throughput method to set up a grid of shapefiles over the field, with each shapefile covering the position of a single field plot. Minor adjustments (typically <0.1 m) to the placement of cells within these grids were made, if necessary, to ensure that the cells aligned with the plot boundaries. Percent canopy area, mean canopy elevation, maximum canopy elevation, and median soil elevation for each plot were then downloaded into the shapefile attribute tables, using the zonal statistics plugin.

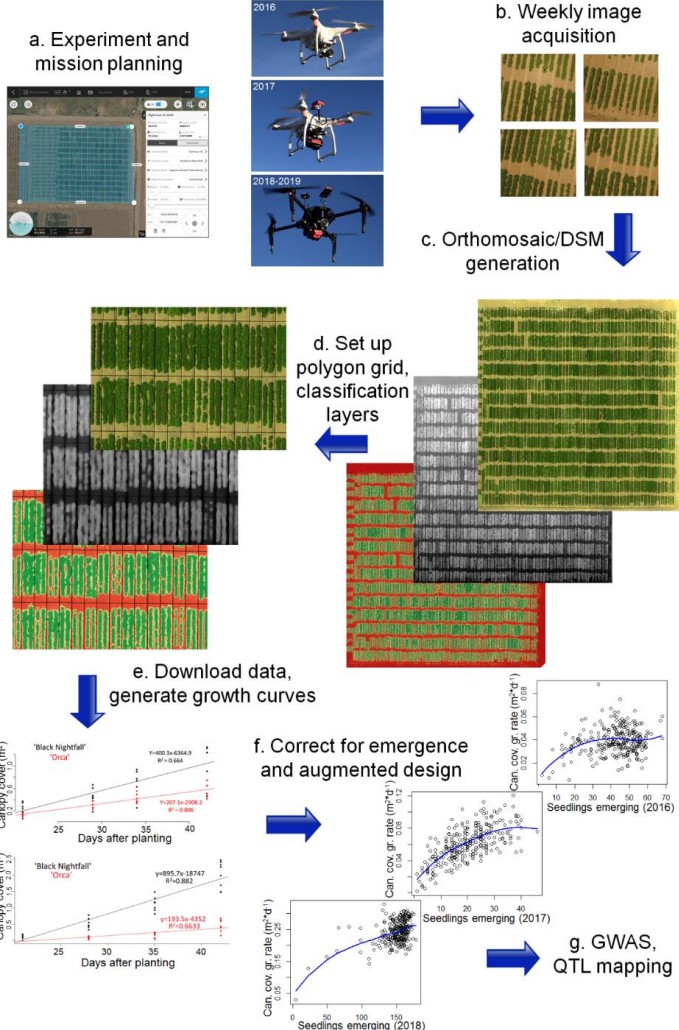

**Figure 1.** Phenotyping pipeline. (**a**) Experiment and mission planning. Flights were conducted using Map Pilot (2016), Atlas Flight (2017), and DJI Ground Station Pro (2018–2019), based on hardware compatibility. Aircraft and cameras included a DJI Phantom 3 Pro with stock RGB camera (2016), a DJI Phantom 3 Pro with a Parrot Sequoia multispectral camera (2017), and a DJI Matrice 100 with a RedEdge-M (2018–2019). (**b**) Field images were collected at 21, 28, 35, and 42 days after planting (DAP). (**c**) Orthomosaics and digital surface models were generated from raw images using Pix4Dmapper Pro. Vegetation indices, spatial classification layers, and masked orthomosaics were generated using the QGIS raster calculator function. (**d**) Shapefile grids were developed using the QGIS vector grid function. (**e**) Phenotype data were downloaded for each plot, which were used to develop the growth curves. (**f**) Corrections for the number of plants emerging were conducted by taking residuals of a LOESS regression between the growth rate and emergence for each plot. A correction to account for field spatial variation was carried out based on the replicated controls planted throughout each block of each field (augmented design). (**g**) Growth and canopy data were used as phenotype inputs for genetic mapping, through genome-wide association studies (GWAS) and quantitative trait locus (QTL) mapping.

**Table 1.** UAV phenotyping information.

| Year of Trial | Population Evaluated | Aircraft | Primary Camera | Camera Type | Altitude | Mission Planning Application | GSD |
|---|---|---|---|---|---|---|---|
| 2016 | MDP | DJI Phantom 3 Professional | Stock (Sony EXMOR 1/2.3 sensor) | RGB | 17 m | Map Pilot | 0.7 cm/px |
| 2017 * | MDP, BxO | DJI Phantom 3 Professional | Parrot Sequoia | Multi-spectral | 20 m | Atlas Flight | 1.9 cm/px |
| 2018 | MDP, BxO | DJI Matrice 100 | Micasense RedEdge-M | Multi-spectral | 20 m | DJI GS Pro | 1.4 cm/px |
| 2019 | BxO | DJI Matrice 100 | Micasense RedEdge-M | Multi-spectral | 20 m | DJI GS Pro | 1.4 cm/px |

* The first flight of 2017 (21 days after planting) used the materials and methods of 2016.

Attribute table data from the zonal statistics plugin were downloaded for final preparation. Bare soil separated each plot, so the elevations of the plant canopies in a plot were compared to the elevations of the soil immediately surrounding the same plants on the same flight dates, to determine canopy height. Mean and maximum canopy heights were generated by subtracting the median sDSM value for a plot (corresponding to the top of the soil bed) from the mean or maximum cDSM value of the same plot at the same time-point. Although DSMs had sufficient precision to distinguish plants at even the earliest sampling points, the vertical accuracy of orthomosaics between the time-points was not sufficient to be useful (± several meters), so all comparisons were made within the sampling time-points and within a cell in the grid of shapefiles. QTL mapping and genome-wide association studies (GWAS) of canopy area-related traits were run with a correction for spatial variation by augmented design. This method was based on comparisons with a set of control varieties that were planted in each replication block, throughout the field. The correction for augmented design was not used for the height-related traits, where it typically reduced data quality, and the significance of results due to low variation in height due to spatial factors. Emergence rate had a major effect on many growth rate metrics and was variable among varieties and years. Variation in emergence was corrected in two ways. First, the number of hand-counted emerging plants was used to evaluate the average growth on a per-plant basis. This correction was relatively conservative, as plants in plots with low emergence rates faced reduced competition from neighbors and the corresponding plots often display greater per-plant growth rates. The main emergence-correction method employed a locally weighted LOESS regression [38] between the emergence and the growth rate for the plots of each population and year. The residuals of each field plot from the LOESS curve was used as an emergence-corrected phenotype for the GWAS and QTL mapping.

The main UAV-based phenotyping methods were complemented and validated by alternate forms of data collection (Figure 2). Canopy heights of plants at 42 days were taken for the MDP from 2016–2018, and canopy heights of the BxO population were taken at 42 and 63 days after planting in 2018 and 2019. Direct comparisons involving Black Nightfall and Orca were made by student's *t*-test, with a significance threshold of 0.05. The ability to precisely quantify the canopy cover of plant stands with complex shapes was expected to be much greater by the UAV-based methods than by the hand measurements or human visual evaluation. Therefore, to validate our main UAV-based canopy coverage estimates, the backup RGB data was used from the same flight date. This image set was collected using a different camera (DJI Zenmuse X3), with a different GPS. All data processing between camera types was conducted independently. Soil and canopies were distinguished using an ExG threshold of 0.1, which corresponded with a human visual inspection of true canopy outlines, based on the RGB images. The canopy coverage estimates based on the independent methods were then compared by linear regression.

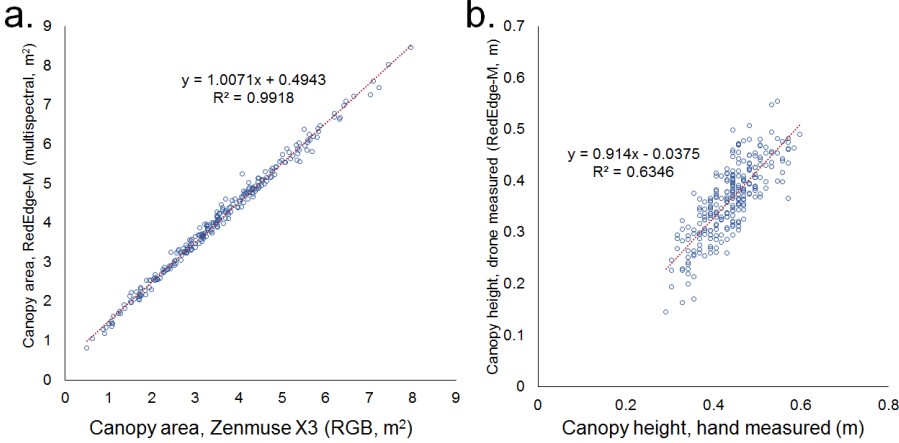

**Figure 2.** Comparison and validation of phenotyping methods. (**a**). Canopy coverage data between the Zenmuse X3 RGB camera and the RedEdge-M multispectral camera were highly correlated, indicating high precision in methodology. The Zenmuse Z3 was used to take true color imagery, which was calculated into the Excess Green (ExG) vegetation index. An ExG value of 0.1 was used as a threshold to distinguish plant area from non-plant area in the field. This classification layer was visually checked against the true color imagery to ensure that it accurately represented the outline of plants. The canopy area collected with the RedEdge-M multispectral camera was processed into NDVI and a threshold of 0.5 was used to distinguish plant from non-plant. Cameras were flown over the same field on the same day, and all data processing was independent. (**b**). Hand-measured canopy height was correlated with that measured by UAV, although the relationship is not as strong as that of the area measurements. UAV canopy heights were determined by subtracting the median soil level of a plot from the maximum canopy surface level. Both data sets were taken in the BxO population at 42 DAP in 2019.

## 2.3. Genotyping and Genetic Mapping

The MDP was genotyped using the Illumina BARCBean6K_3 BeadChip and Genotyping-by-Sequencing (GBS), as described previously [27]. The SNP data consisted of 211,763 markers. The data set was filtered to remove varieties not included in the original 280-member panel, and all SNPs with a minor allele frequency below 0.1 in this group were eliminated, leaving 81,337 final SNPs. Genotype data for the MDP is available online [39]

GWAS of the MDP was conducted by the generalized linear model in TASSEL [40] through the SNiPlay interface [41]. A correction for population structure was performed using the first five principal components of the genetic data. The Manhattan plots were visualized using the qqman R package [42]. A Bonferroni significance threshold was assigned based on a false-discovery rate of 0.05 (logarithm of the odds (LOD), or $-\log_{10}(p) = 6.21$) and was included in all Manhattan plots.

In the BxO population, DNA was extracted from the individual F8 seeds, using a modified CTAB protocol. Quality was checked by gel and NanoDrop spectrophotometer, and the population was genotyped using the Illumina BARCBean6K_3 BeadChip [43]. A total of 1164 SNPs was segregated between the parents and among the RILs. Linkage mapping was conducted in the ASMap R package [44], with a minimum linkage significance threshold of $10^{-5}$ between markers. Composite interval mapping was performed using maximum likelihood, through the EM algorithm in R/qtl [45,46]. The 95th percentile of LOD scores of 1000 randomized permutations of the data (LOD = 3.03) was used as a significance threshold.

## 3. Results

### 3.1. Growth Measurements

Canopy coverage data collected by independent RGB- and multispectral-based methods were highly positively correlated ($R^2 = 0.99$, Figure 2a). The slope of the relationship was also very close

to one, with the two forms of data collection giving similar numeric values. Similarly, the height data generated by alternate hand-measured and UAV collected methods were positively correlated ($R^2 = 0.63$, Figure 2b). The relationship between these was not as close as those of the canopy area methods. Growth rate measurements between weeks were also highly correlated, with the correlation coefficients at times exceeding 0.9 (Figure 3). Correlations were stronger between weekly data collection of the canopy area (Figure 3a–c) than those of canopy height (Figure 3d–f). Canopy volume measurements were intermediate in correlation (Figure 3g–i). Differences in these correlations also existed based on allelic variation. Individuals with the lodging-associated allele S07_34512442_G (see Results Section 3.2), for example, had relatively low correlations in height between 35 DAP and 42 DAP in the MDP in 2018 ($R^2 = 0.33$). In contrast, individuals with the alternate allele, associated with lodging resistance, had a much higher predictability in height in the same trial ($R^2 = 0.73$, Figure 3f).

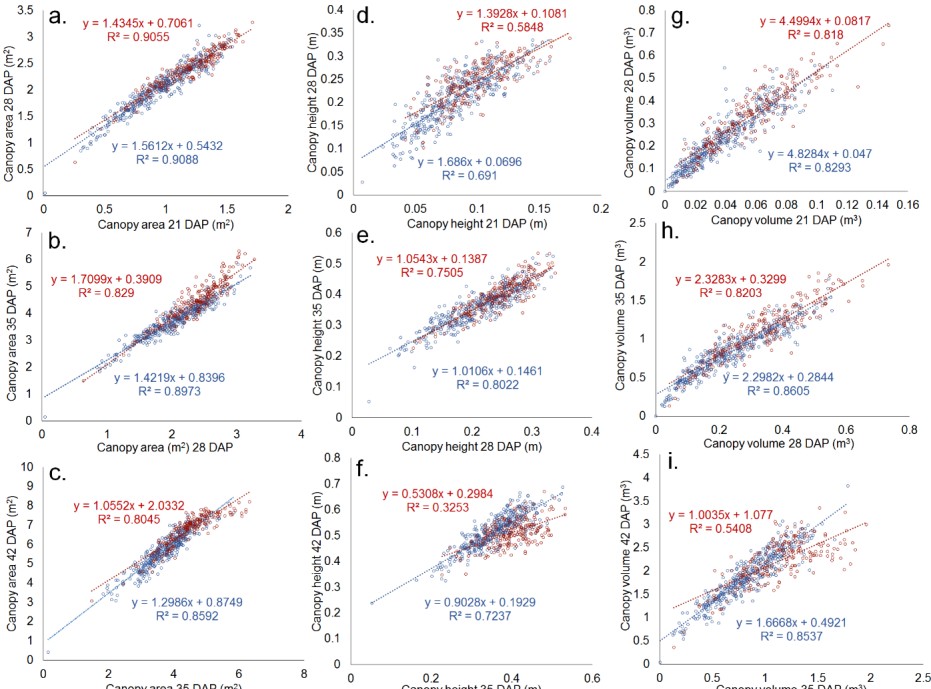

**Figure 3.** High-precision drone-based evaluation of canopy area, height, and volume leads to strong data reproducibility between weeks. SNP S07_34512442 on Pv07 was related to canopy area ($m^2$, (**a–c**)), height (m, (**d–f**)), and volume ($m^3$, (**g–i**)) in the Middle American Diversity Panel (MDP) in 2018. Types with a thymine at this position (red) initially have a greater canopy area and height than those with a guanine (blue) (**a,d,g**). By 42 DAP, varieties with a thymine at the position experience greater lodging, reducing the rate of canopy height (**f**) and volume increase (**i**).

### 3.2. Genome-Wide Association Studies

GWAS identified numerous loci associated with growth rate (Table S1), including at least two related to emergence-corrected canopy cover growth rate in the MDP (Figure 4). One of these, on chromosome Pv01, was identified in 2016 when grown on organically managed land. A second locus, on Pv11 (LOD=6.19), approached the Bonferroni-corrected significance threshold (LOD = 6.21). In 2018, a highly significant locus on Pv07 was identified when the population was grown in a conventional system. The most significant SNP (S07_34512442) from this GWAS was located at Pv07 position 34,512,442. GWAS of the emergence rate failed to identify any significant SNPs in the MDP, although a locus on Pv06 was nearly significant (LOD = 6.19) in 2018 (Figure S2). GWAS of canopy height based on hand-measurements and drone-based methods, each identified marker-trait associations. In 2017, a peak on chromosome Pv01 in the vicinity of *PvTFL1y* was related to a hand-measured height, while in 2018, loci on Pv07 and Pv08 played a significant role (Figure S3). Drone-based height measurements identified

a single SNP on Pv03 and a region of Pv05 that were related to the trait in 2017. In 2018, SNPs on Pv04 and Pv08 were related to canopy height (Figure 5).

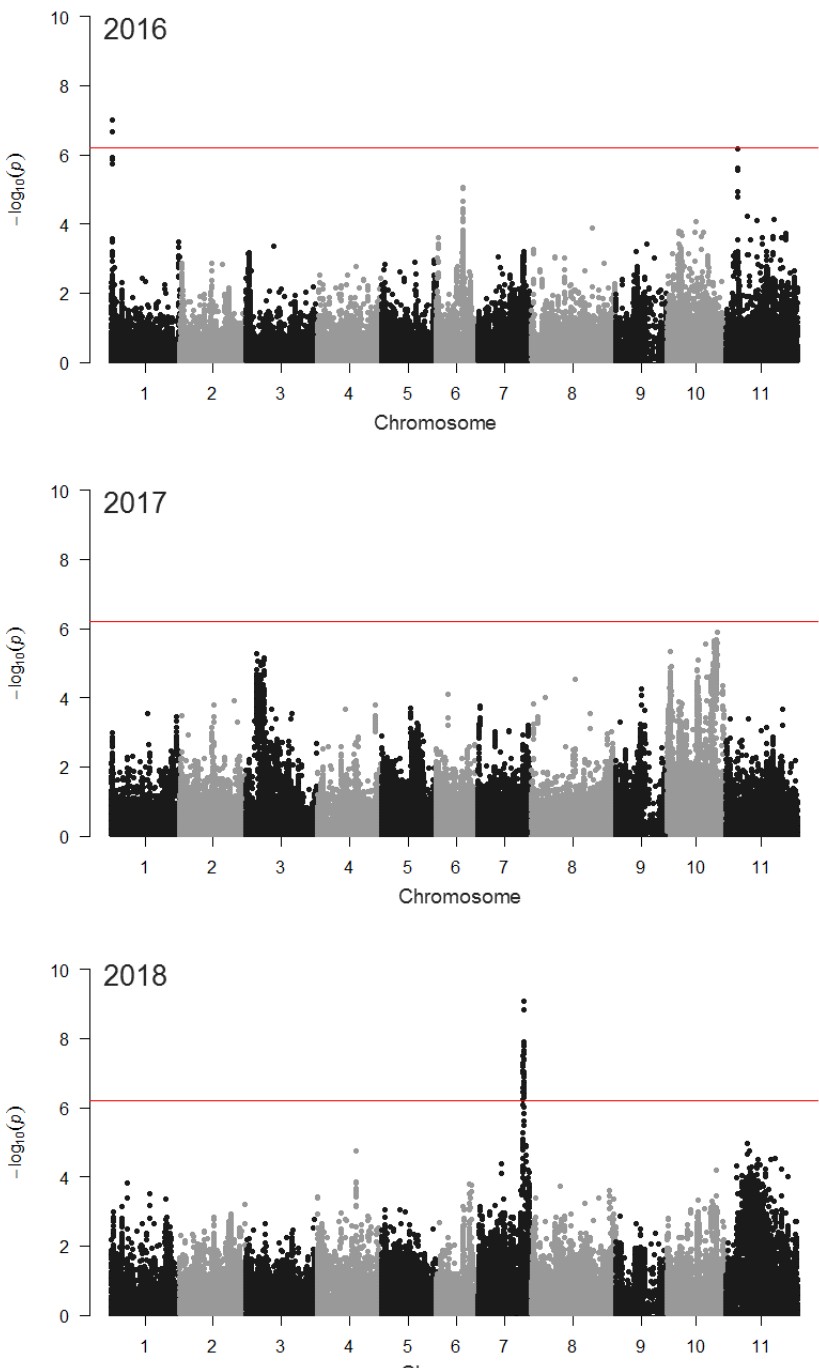

**Figure 4.** MDP canopy cover growth rate 21–42 DAP, emergence correction by LOESS residuals. Significant SNPs were found on different chromosomes in each year, reflecting the complex interactions between genotype and environment for canopy cover growth rate. By far the most significant SNPs from any year were identified in 2018. The most significant SNP (S07_34512442; Pv07 position 34,512,442) was found in the 5′ UTR of Phvul.007G221800, a gene model that was postulated to pleiotropically affect canopy architecture, height, and lodging [27]. GWAS was conducted as a general linear model in TASSEL with a 81K SNP set, correction for emergence by residuals from a LOESS model of growth rate, as function of emergence. The red lines indicate a Bonferroni-corrected significance threshold ($-\log_{10}(p)$), or logarithm of the odds (LOD) = 6.21). See Table S1 for more information on significant SNPs.

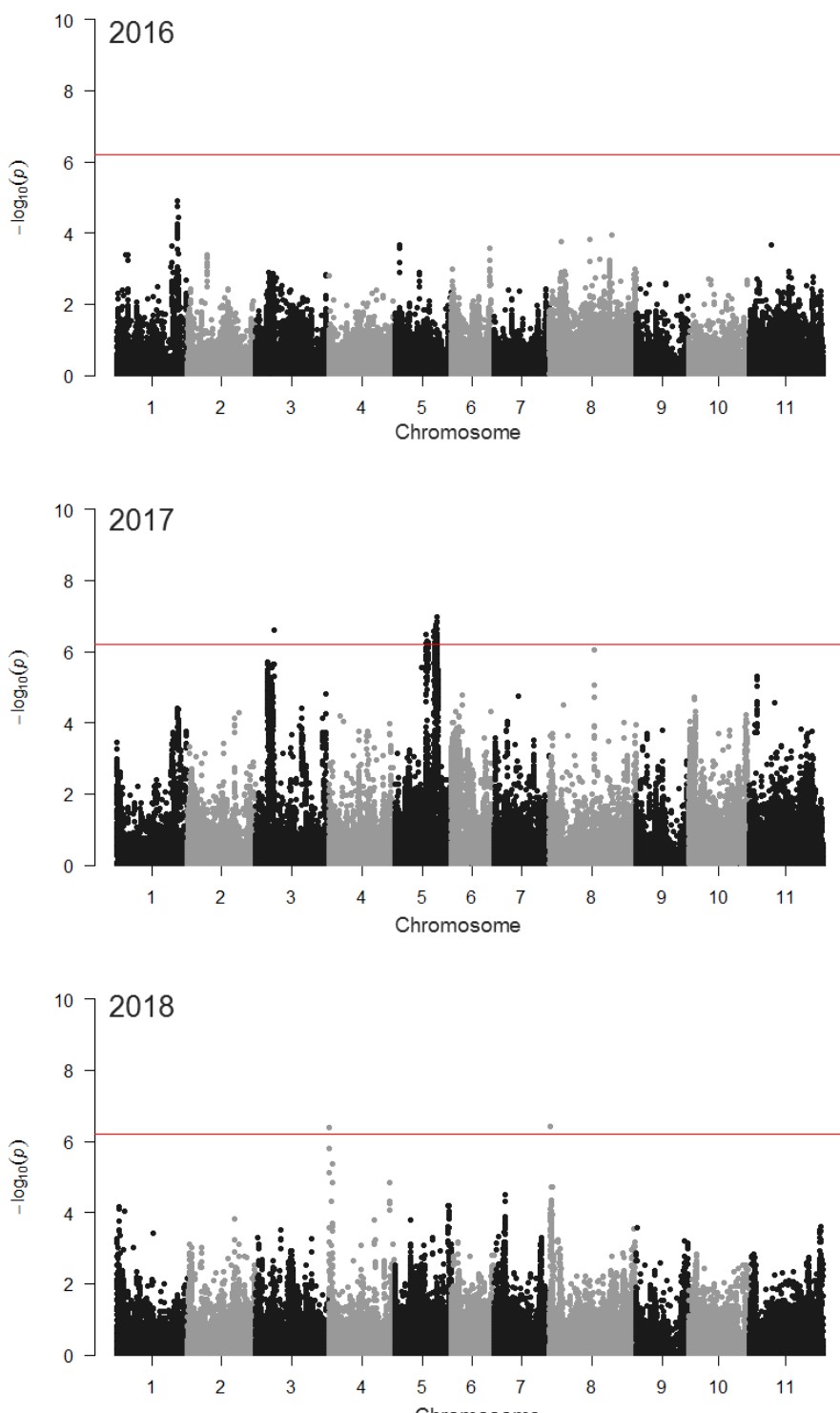

**Figure 5.** Drone-measured maximum canopy height GWAS. The most significant SNPs in 2016 were clustered near *PvTFL1y* on chromosome Pv01. The most significant SNPs in 2018 are found on Pv04, near a canopy height QTL identified by Moghaddam et al. [27]. No correction for augmented design was implemented. GWAS was conducted as general linear model in TASSEL with a 81K SNP set. The red lines indicate a Bonferroni-corrected significance threshold (LOD = 6.21). See Table S1 for more information on significant SNPs.

### 3.3. Biparental Population Phenotyping

Black Nightfall and Orca displayed statistically significant differences in several traits in the 2016–2019 trials (Table S2, Figure S4). During the 21–42 DAP interval, Black Nightfall had a faster canopy coverage rate and simultaneously grew taller than Orca. The daily drone-measured increase in canopy area for plots of Black Nightfall surpassed those of Orca by 70% in 2016, 366% in 2017, 72% in 2018, and 247% in 2019. This difference was statistically significant in each year. During the same interval, Black Nightfall also grew taller in hand-measured canopy height, outcompeting Orca in each year from 2016–2019 by 36%, 26%, 18%, and 50%, respectively. These differences were also statistically significant in each year. Drone-based height measurement also identified differences between Black Nightfall and Orca in 2017–2019, when multispectral imagery was used for DSM construction. Differences in the percentage of seeds that successfully germinated and emerged could have a strong influence on canopy-cover growth rate. In each year, other than 2016, stand counts indicated that Black Nightfall had a significantly higher percentage of seedlings emerging than Orca. To correct for this, the average canopy cover growth rate per emerging plant was compared between varieties. In 2016, 2017, and 2019, Black Nightfall had a statistically significantly faster growth rate per plant than Orca, despite having a greater between-plant competition due to its higher emergence rate. In 2018, Black Nightfall again had a greater average per-plant canopy cover growth than Orca, although the difference was not significant ($p = 0.06$).

### 3.4. Linkage Map Development and QTL Analysis

A linkage map consisting of 730 SNPs was used to identify the marker-trait associations in the BxO population (Figure S5). This map consisted of 11 linkage groups that were arranged to match the numbers and direction of the chromosomes agreed upon by the *Phaseolus* community [47,48]). The linkage map was comparable in size to others in the species at 1138 cM. Individual chromosomes ranged from 64 cM to 132 cM, with between 41 and 109 markers. Average distance between markers was 1.6 cM, with a maximum distance of 26 cM.

QTL mapping in the population identified the regions associated with drone-measured canopy height, hand-measured canopy height, and canopy cover growth rate, using a residual from a LOESS model as an emergence correction (Table S3). QTL mapping identified three loci associated with canopy cover growth rate. The most significant of these was found on chromosome Pv07 in 2018 and 2019 (Figure 6). A locus on Pv06 was related to the trait in 2017 and 2018, while a locus on Pv09 had a significant association in 2019. Average seedling emergence was low in 2017 (32%) and 2019 (61%), but relatively high in 2018 (85%). QTL mapping for emergence rate showed that a major locus on Pv10 was the primary regulator of growth rate in 2017 and 2019, when emergence was low. In contrast, in the more favorable conditions of 2018, variation in emergence was related to a locus on chromosome Pv07 (Figure S6).

Several QTLs were related to plant height in the BxO population. In all three years, chromosome Pv09 was significantly related to drone-measured maximum canopy height at 42 DAP (Figure 7). In 2017 and 2019, another QTL was identified with a significant role on Pv10. A major QTL for hand-measured height 42 DAP was identified on Pv07 in 2018, while a secondary locus on Pv01 was also weakly significant (Figure S7). Lodging scores taken simultaneously identified the same QTL, indicating that the variation in canopy height was likely the result of changes in canopy architecture. Hand-measurements of canopy height did not find a significant QTL on Pv07 at 42 DAP in 2019, while QTLs on Pv09 and Pv10 contained significant QTLs (Figure S7). By 63 DAP in 2019, hand-measured canopy height measurements indicated that Pv07 was significantly associated with height in the population, while the role of the other chromosomes was reduced (Figure S7).

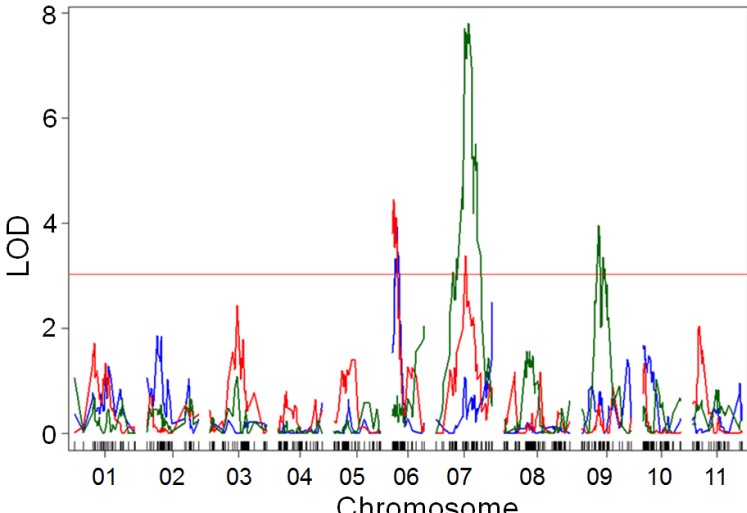

**Figure 6.** Quantitative trait locus (QTL) mapping of canopy area growth rate in the BxO population in 2017 (blue), 2018 (red), and 2019 (green), with LOESS residuals used as correction for emergence. Major QTLs were found on chromosomes Pv06, Pv07, and Pv09. Growth rate was strongly influenced by environment, leading to different QTLs controlling the trait among years. The chromosome Pv07 QTL was found on the same chromosome arm as the most significant SNP from the GWAS of the MDP. The underlying gene might be pleiotropically related to canopy architecture, height, and lodging. Plots were four-fold larger in 2019 than in previous years, potentially leading to improved QTL resolution. Results based on slope of canopy cover increased between 21, 28, 35, and 42 days after planting, with a residual-based correction for a LOESS model of emergence rate and a correction for augmented design. The 95th percentile of LOD scores from 1000 randomized permutations of the data (LOD = 3.03) was used as a significance threshold.

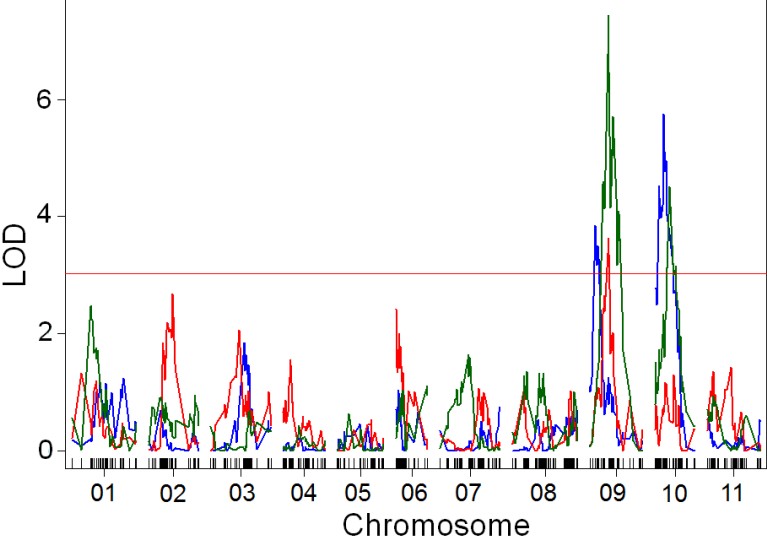

**Figure 7.** QTL mapping of drone-measured maximum canopy height 42 days after planting in 2017 (blue), 2018 (red), and 2019 (green). QTLs on Pv09 are related to maximum canopy height in all three years, whereas QTLs on Pv10 are responsible for the trait in two years. The same Pv09 SNPs were most significant in 2018 and 2019. No correction for augmented design was implemented. The 95th percentile of LOD scores from 1000 randomized permutations of the data (LOD = 3.03) was used as a significance threshold.

## 4. Discussion

The QTLs identified in this work will be useful for improving the weed suppression of common bean through faster canopy development. The most significant SNP from GWAS of the MDP was associated with canopy growth rate and was located at position 34,512,442 on Pv07, approximately 3 kb from a SNP identified by Moghaddam et al. [27] at position 34,509,509, in their evaluation of the canopy architecture. The SNP identified by Moghaddam et al. was predicted to cause a non-synonymous S740T mutation in the CDS of Phvul.007G221800. The SNP most strongly related to the MDP canopy growth rate in this study was in the 5′ UTR of the same gene model. Phvul.007G221800 was predicted to encode a leucine-rich repeat receptor-like protein kinase. Closely related members of this gene family included a BRASSINOSTEROID INSENSITIVE 1 precursor (transcript 29904.m003011) in *Ricinus communis* L. [27]. BRASSINOSTEROID INSENSITIVE 1 proteins are membrane-bound receptors of brassinolides, which regulate plant growth. Mutations in these genes cause severe growth stunting when mutated in *Arabidopsis thaliana* (L.) Heynh. [49]. In common bean, this locus pleiotropically regulates several characteristics, possibly by controlling plant stem length and flexuosity, which contribute to lodging.

This result was consistent with our observations, which showed that in individuals with the one SNP variant associated with taller plants, canopy height and volume increases at a predictable rate between 35 and 42 DAP. In contrast, lodging began to occur during this interval in individuals with the alternative SNP allele. This led to less predictable heights and volumes between these sampling points, and a less linear relationship between weeks. This most significant Pv07 growth rate SNP from the MDP was also within 800 kb of *PvTFL1z*, a locus believed to control determinacy in Michigan-type small, white-seeded beans (Phvul.007G229300, [32,50,51]). Identifying the polymorphisms responsible for this variation could be useful for improving the canopy characteristics of the common bean. Loci controlling seedling emergence, growth rate, and height on other chromosomes would also be strong candidates for continued genetic evaluation. These detailed molecular characterizations will parallel recent studies on other traits in the species, such as determinacy [31], seed pigmentation [52], photoperiod sensitivity [53], and pod shattering [54–56].

Drone- and hand-measurement of plant height, each yielded significant QTLs in the BxO population, but their loci and significance differed (Figure 7, Figure S7). The differences between drone- and hand-height measurement might be partially due to subtle differences in the precise trait being measured. Hand-based height measurements took the height from the soil to the typical canopy level of a plot, whereas drone-based canopy heights measured the distance from the soil to the highest position of the digital surface model in the plot. While these models typically did not have sufficient resolution to capture individual guides (runners), they were much more sensitive to the individual tall plants. If most but not all plants in a plot had lodged, the hand-measured height would be reduced, while the drone-based height measurement would continue to measure the height of the tallest plants. This could explain why the drone-based maximum height measurement methods did not identify the lodging locus on Pv07, but more consistently identified the loci on Pv09 and Pv10. Drone-based methods identified highly significant QTLs for maximum canopy height on Pv09 in all three years, providing strong evidence that this chromosome was involved in the variation in plant height between Orca and Black Nightfall. Similarly, Pv10 included maximum height QTLs in two of three years. These chromosomes did not have significant QTLs when the same techniques were applied in the MDP, indicating that some of the responsible alleles might be at low frequency among the Middle American beans, or might have complex interactions with other loci. In agreement with the drone-measured results, manual height measurements identified loci on Pv09 and Pv10 in 2019. In 2018, the major locus controlling hand-measured height was on Pv07, the chromosome associated with lodging and canopy area growth rate.

Plant growth is a complex phenomenon, influenced by interactions between numerous genetic and environmental factors. This phenotypic complexity requires multiple years of field evaluations to effectively identify loci that regulate traits of interest. Our results indicated that many factors, including emergence rate, genotype, and environmental differences interact to produce a range of growth-related traits.

The role of some chromosomes and QTLs are fairly robust and can be identified in multiple populations and years. The role of other QTLs might only be of major importance in a narrower combination of environmental conditions and genetic backgrounds. In contrast to the relatively stable effects of the Pv07 locus found by Moghaddam et al. [27] and in this study, Resende et al. [57] did not identify any loci on Pv07 associated with canopy architecture. In their population, which consisted of 115 Middle American, 66 Andean, and seven admixed types, major lodging and architecture QTLs on Pv01 (possibly related to *PvTFL1y*, [31,50]) and Pv08 controlled trait variation. This highlights the importance of evaluating germplasm across a broad array of locations and environments.

Our results validate the utility of UAV in identifying QTLs in common bean. These are the first QTLs identified in this species using UAV-based imagery, although UAVs were used for other purposes in the species [58–61]. Use of UAVs to identify genetic variation will likely grow in the future, as a complement to the existing phenotyping methods. The strong correlation between multispectral and lower-cost RGB imagery indicated that both methods were highly effective for evaluating canopy cover, and this utility could potentially extend to a variety of other traits. This is a promising result, particularly for research programs with limited budgets.

Further advances in UAV-related hardware and software would be of central importance for future crop phenotyping. Improved sensor spectral resolution, photogrammetric algorithms, and data extraction tools would be of central importance for improving UAV-based phenotyping. In particular, the improvement of photogrammetric tools might be of particular importance for improvement of estimations of plant height. Existing photogrammetric tools are much better suited to reconstructing details within the plane of view (i.e., the length and width of objects, when measured from above) relative to the perpendicular axis (i.e., the height of the same objects). While the current toolkit allows us to create 2D vegetation index orthomosaics with a centimeter-level resolution and high spatial fidelity (< 1% error), the quality of DSM values is comparatively low on the scales studied (>10% error). While these are still useful as relative measurements between objects, the absolute values of these height measurements on plant canopies less than a meter tall, as described here, are still lacking. Improvements in photogrammetry and computational power are likely to greatly reduce these issues in the near future. Similarly, reductions in the cost and weight of high-precision lidar units would offer another major solution to these issues, in coming years. In the meantime, researchers can apply the existing tools to maximize their data quality through sound data collection protocols, such as flying at relatively low altitudes, using high image overlap, collecting imagery during optimal weather conditions, and ground truthing traits when necessary. The use of UAVs and other remote sensing tools in agriculture is likely to expand drastically in the coming years. The remotely identified QTLs described here are likely to be the first of many found in common bean.

Characterizing the loci responsible for vigorous early-season growth is an important task. The identification of genetic regions controlling this trait would allow for the development of cultivars that compete with weeds and use water more efficiently. In turn, this would increase productivity and reduce the weed control required for production of a major crop species. Ultimately, an improved understanding of early growth rate would have repercussions for farm profitability, labor availability, pesticide use, and ecological health for communities around the globe.

## 5. Conclusions

In this work, we developed novel UAV-based methods to study the genetic basis of early growth traits in a major crop species, common bean. These studies were conducted on two major populations, one encapsulating a highly diverse assemblage of crop varieties, and another that was bred ad hoc to display maximum variation in early growth rate. Our results suggest that the regions on chromosomes Pv07, Pv09, and Pv10 were particularly important in regulating these traits. These genotype–phenotype associations were the first to be discovered in common bean, using UAVs or remote-sensing methods. They would be of central importance in breeding new varieties of the species that are less reliant on synthetic herbicides

and which use water more efficiently. This would be critical for adaptation to future environments under changing consumer demands and climate conditions.

**Supplementary Materials:** The following are available online at http://www.mdpi.com/2072-4292/12/11/1748/s1, Figure S1. Layouts of evaluated fields in (a) 2016, (b) 2017, (c) 2018, and (d) 2019. The Middle American Diversity Panel (MDP) was grown 2016–2018, while the Black Nightfall x Orca (BxO) recombinant inbred population was grown 2017–2019. In each year, the field was separated into blocks with replicated controls in each block to account for spatial effects (augmented design, Federer and Raghavarao 1975). MDP blocks are indicated by dark blue boxes, BxO blocks are in light blue. Thin black lines represent plot shapefile boundaries for data extraction. All fields are represented to scale with one another. Figure S2. GWAS of seedling emergence rate in the Middle American Diversity panel. A SNP on Pv06 (LOD = 6.19) approached significance based on a conservative Bonferroni threshold (LOD = 6.21) of 2018 data. GWAS conducted in TASSEL based on a general linear model through the SNiPlay interface. Figure S3. Hand-measured canopy height GWAS in the MDP. In 2017, a significant QTL was identified near PvTFL1y on chromosome Pv01. In 2018, the Pv07 locus associated with growth rate was also associated with hand-measured canopy height. GWAS conducted in TASSEL based on a general linear model through the SNiPlay interface. See Table S3 for more information on significant SNPs. Figure S4. 2018 emergence and growth rate in the Middle American diversity panel, including Black Nightfall and Orca plots as controls. The x axis represents the number of seeds germinated out of 180 seeds planted. Black Nightfall has stronger seedling emergence, and above-average growth rates, even after emergence correction. Orca has lower emergence, and below-average growth rates relative to types with similar emergence rates. Figure S5. Black Nightfall x Orca (BxO) population linkage map: Linkage groups (vertical lines) were ordered and oriented to correspond with the 11 chromosomes of the common bean reference genome [46,47]. A total of 730 markers (horizontal bars) were distributed across 1138cM of recombination space. Figure S6. QTLs related to emergence rate. Robustness of germination and emergence has a profound effect on canopy growth. Emergence was low in 2017 (32%, blue) and 2019 (61%, green), but relatively high in 2018 (85%, red), and different QTLs controlled the trait between these years. The 95th percentile of LOD scores from 1000 randomized permutations of the data (LOD = 3.03) was used as a significance threshold. Figure S7. QTL mapping of hand-measured heights in the BxO population. At 42 DAP in 2018 (red), a major peak on Pv07 was evident. At 42 DAP in 2019 (dark green), several smaller QTLs were related to the trait, but by 63 DAP (light green), the Pv07 QTL had become significant as plants began to lodge. Results were based on data without correction for augmented design. The 95th percentile of LOD scores from 1000 randomized permutations of the data (LOD = 3.03) was used as a significance threshold. Table S1. Positions of SNPs significantly associated with traits through GWAS. Table S2. A comparison of early-season growth between Black Nightfall (BN) and Orca. Table S3. Summary of QTLs significantly associated with growth rate in the BxO RI population. The most significant SNP(s) are listed along with the most significant SNP in the upstream and downstream directions. SNP coordinates are based on v2.1 of the Phaseolus vulgaris Andean genome reference sequence [61]. QTLs with identical positions between traits or years are indicated with asterisks in the chromosome column.

**Author Contributions:** Conceptualization, P.G. and T.A.P.; methodology, T.A.P.; field management, T.A.P. and A.P.; software, T.A.P.; writing—original draft preparation, T.A.P. and P.G.; writing—review and editing, T.A.P. and P.G.; supervision, P.G.; project administration, P.G.; funding acquisition, P.G. and T.A.P. All authors have read and agreed to the published version of the manuscript.

**Funding:** Funding for TAP was provided through a Clif Bar Family Foundation Seed Matters fellowship and Lundberg Family Farm research support, with additional research funding from Western SARE and the UC Davis Plant Sciences Department.

**Acknowledgments:** Brian Cardello provided the UAV for 2016 data collection and provided assistance with remote pilot certification process. Seeds of the ADP and MDP were provided by R. Lee and P. McClean (North Dakota State University). Undergraduates Natalie Hamada, Emily Yang, Ariel Herrera, Jose Pimentel, Matthew Bustamante, Emily White, Julia Gonzales, Paige Augello, Vivian Wu, Nathalie Gonzalez, and Aung Nyein contributed to field management and phenotyping. Jorge C. Berny Mier y Teran offered suggestions related to GWAS, linkage mapping, and QTL mapping. Taylor Nelson and Alex Mandel provided suggestions for geospatial methods; Umair Gull collaborated on developing UAV techniques.

**Conflicts of Interest:** The authors declare no competing interests. The funding sponsors had no role in the design of the study; in the collection, analyses, or interpretation of data; in the writing of the manuscript; or in the decision to publish the results.

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
