# Peer review of "Determining the Genetic Control of Common Bean Early-Growth Rate Using Unmanned Aerial Vehicles"

_remotesensing, doi:10.3390/rs12111748_

Round 1

Reviewer 1 Report

This paper examined the drone-based evaluation of canopy area, height, and volume, which is interesting but needs explanations about the validation statistics using in-situ measurements.

*What the QTL stand for? quantitative trait locus?

*Fig 2. The canopy area, height, and volume according to DAPs look closely correlated. But (f) and (g) in red (thymine) showed lower correlations when using a fitting line. You can try a logistic or exponential curve for fitting them, which may increase the R-squared values.

*You presented the validation statistics using in-situ measurements only for canopy height. Can you present the validation statistics for canopy area and volume? Or you should explain about that in more detail. 

*Conclusion sections will be necessary.

Reviewer 2 Report

The research is interesting and, in my opinion, with a high interest and potential. However, the methodology is to be improved. As the authors honestly described, there are differences in correlating QTLs with phenotypes, when different measurements methods - UAV and manual - are used.

The good part is that some discussion on the subject is included. But, to what is to be understood from Table S2, there are major differences in canopy height when measured by hand and by UAV in each of the years - 13.1 compared to 38.31, 45.0 compared to 50.80, and a correction factor does not seem feasible to be introduced.

Furthermore, was canopy growth rate, for instance, compared with values obtained by other methods than UAV imaging?

I would suggest a more detailed discussion on what can be improved, not necessarily in this study, but in measuring/imaging methods in general, on correlating phenotypes with QTLs.

Reviewer 3 Report

The authors evaluate early season growth vigor of the common bean using both UAV and ground based measurements. Proxies of growth data, namely canopy cover and height derived from NDVI and the excess green VI were used to find QTLs associated with growth parameters/traits. The authors suggest that UAVs provide a high throughput phenotyping solution. This is a novel study which validates the broad scope of UAV applications in the field of high throughput phenotyping. However, the manuscript has several low quality, pixelated figures with unreadable captions.
Line 13: Please spell out QTL at first mention. Most readers of this journal will not be acquainted with quantitative trait locus and other plant genetics terminology.
Lines 82-84: Please include a map of the experimental design of plots in supplementary Information or reference another study here such maps were displayed.
Line 105: Please re-save this figure at higher resolution and increase the font size of the y-axis labels which are unreadable.
Line 123: Given that your GSD is 1cm and that you are conducting multi-year, weekly overflights using different UAVs, there is no mention of ground control points obtained using a survey grade GPS. What is the horizontal/vertical accuracy of this study. This is important as it related to the canopy height derived from the DSM.
Line 188: I'm assuming DAP stands for "days after planting" if so, please spell out at first mention.
Line 185: Please increase the font size of the y-axis labels which are unreadable. Add units in parenthesis to variables in caption. Captions need to be standalone so that the reader/reviewers do not have to guess.
Line 261: Please increase the font size of the y-axis labels which are unreadable. Please spell out LODscore at first mention. Captions need to be standalone so that the reader/reviewers do not have to guess.
